# microRNAs Regulating Human and Mouse Naïve Pluripotency

**DOI:** 10.3390/ijms20235864

**Published:** 2019-11-22

**Authors:** Yuliang Wang, Abdiasis M. Hussein, Logeshwaran Somasundaram, Rithika Sankar, Damien Detraux, Julie Mathieu, Hannele Ruohola-Baker

**Affiliations:** 1Paul G. Allen School of Computer Science & Engineering, University of Washington, Seattle, WA 98195, USA; 2Institute for Stem Cell and Regenerative Medicine, University of Washington, Seattle, WA 98109, USA; asishussein1@gmail.com (A.M.H.); logeshsomasundaram@gmail.com (L.S.); rithikashankar100@gmail.com (R.S.); damiendetraux@hotmail.com (D.D.); 3Department of Biochemistry, University of Washington, Seattle, WA 98195, USA; 4Department of Comparative Medicine, University of Washington, Seattle, WA 98195, USA

**Keywords:** microRNA, naïve and primed pluripotent stem cells, embryonic diapause, shh

## Abstract

microRNAs are ~22bp nucleotide non-coding RNAs that play important roles in the post-transcriptional regulation of gene expression. Many studies have established that microRNAs are important for cell fate choices, including the naïve to primed pluripotency state transitions, and their intermediate state, the developmentally suspended diapause state in early development. However, the full extent of microRNAs associated with these stage transitions in human and mouse remain under-explored. By meta-analysis of microRNA-seq, RNA-seq, and metabolomics datasets from human and mouse, we found a set of microRNAs, and importantly, their experimentally validated target genes that show consistent changes in naïve to primed transitions (microRNA up, target genes down, or vice versa). The targets of these microRNAs regulate developmental pathways (e.g., the Hedgehog-pathway), primary cilium, and remodeling of metabolic processes (oxidative phosphorylation, fatty acid metabolism, and amino acid transport) during the transition. Importantly, we identified 115 microRNAs that significantly change in the same direction in naïve to primed transitions in both human and mouse, many of which are novel candidate regulators of pluripotency. Furthermore, we identified 38 microRNAs and 274 target genes that may be involved in diapause, where embryonic development is temporarily suspended prior to implantation to uterus. The upregulated target genes suggest that microRNAs activate stress response in the diapause stage. In conclusion, we provide a comprehensive resource of microRNAs and their target genes involved in naïve to primed transition and in the paused intermediate, the embryonic diapause stage.

## 1. Introduction

MicroRNAs are highly conserved small non coding RNA that post-transcriptionally regulate their target genes by binding mainly to the 3′UTR region of protein-coding mRNA, resulting in translational repression or transcript destabilization [1]. MicroRNAs have been shown to be essential for animal development [2,3], which is not surprising since they play key roles in various cellular processes such as cell cycle progression, proliferation, apoptosis, and differentiation [1,4,5]. MicroRNAs can regulate a large fraction of the transcriptome and function as rapid switches by regulating gene expression in a coordinated, fine-tuned manner. Indeed, a single microRNA can target hundreds of genes at the same time, and multiple microRNAs can target the same gene. MicroRNAs have therefore emerged as major regulators of cell fate determination, from maintenance of stem cell self-renewal to commitment toward specific lineages [6,7,8]. Disruption of enzymes involved in microRNA biogenesis (Drosha, DGCR8, Dicer) leads to the failure of embryonic stem cells (ESC) to downregulate pluripotency markers and to differentiate after exposure to a differentiation signals [9,10,11,12,13,14]. In addition, microRNAs are required for the generation of induced pluripotent stem cells (iPSC) [15] and regeneration in general, including heart regeneration and its opposing process, maturation [16,17,18,19,20].

We and others have shown that pluripotent stem cells express a very specific set of microRNAs [7,21,22,23,24], including miR-302 cluster, miR-372 (miR-290-295 cluster in mouse), the let-7 family and the primate specific imprinted CMC19 cluster containing the miR-520 family. Some of these microRNAs, named Embryonic Stem cell specific Cell Cycle regulating miRNA (ESCC miRNA) control the proliferation of ESC by repressing genes involved in cell cycle regulation [12,14,25]. In order to sustain their proliferation rate, pluripotent stem cells rely on glycolysis as their main source of energy [26,27,28]. The ESCC miRNA have also been shown to promote glycolysis in mESC by regulating the Mbd2-Myc-glycolytic enzymes axis [29].

Mouse and human ESC have been stabilized in culture at different pluripotency states corresponding to pre- and post-implantation stages *in vivo* and are referred as naïve and primed cells, respectively [30,31,32] (Figure 1A). Even though these cells are close in a developmental timeline, they are very different in terms of signaling requirements, gene expression, epigenetic landscape, and metabolic signature [26,30,31,32]. In the past few years it has become clear that pluripotency is a very dynamic stage and cells progress through a continuum of pluripotent states with unique properties for each state [30,33,34]. The pre-to-post-implantation transition can be suspended under certain conditions, and this stage is called diapause [35] (Figure 1A). Let-7 has been previously shown to be a potential regulator of diapause [36,37]. Additional microRNA regulators of diapause and their target genes remain under-explored. 

Several microRNA-seq studies have revealed distinct microRNA signatures in naïve and primed mESC [38,39,40], and *Dgcr8* KO experiments have shown that microRNA are essential for the transition from naïve mESC to primed mEpiSC [40]. In particular, the miR-302 cluster is expressed at higher levels in mEpiSC compared to mESC [22,38,40] and facilitates the exit of naive pluripotency in part by promoting the activity of MEK pathway [38]. To our knowledge, no study has compared the expression of microRNAs in naïve and primed human pluripotent stem cells. However, low concentration of the HDAC inhibitor sodium butyrate, which induces primed hESC to de-differentiate to an earlier stage in development [41], increases expression of miR-302 cluster while decreasing expression of miR-372 cluster [22], suggesting common microRNAs are involved in mouse and human naïve-to-primed transition. In this paper we compared naïve hESC grown in 2iLIF media [26,27,42,43,44] with primed H1 for their microRNA profile and analyzed it in parallel with their transcriptomic and metabolomic profiles. In addition, we combined existing datasets in mouse pluripotent cells [38,39,40] in order to find microRNAs regulating important pathways during the naïve to primed transition, and in naïve and primed states. We also identified 38 microRNAs as potential regulators of diapause by combining existing microRNA expression data [37] with our RNA-seq of diapause and post-implantation embryos [35].

We found 2184 consistent microRNA-target gene connections between 280 microRNAs and 647 target genes in human, and 435 consistent microRNA-target gene interactions between 80 microRNAs and 241 target genes in mouse. Importantly, we identified 115 microRNAs that significantly change in the same direction in naïve to primed transition in both human and mouse, many of which have not been previously reported, and serve as a resource for future studies. These microRNAs and their target genes regulate developmental (e.g., Hedgehog pathway) and metabolic pathways (e.g., fatty acid oxidation, OXPHOS) important for pluripotency. Interestingly, we found that microRNAs are likely to repress Sonic Hedgehog (shh) activity in human pluripotent cells. Indeed, microRNAs could down-regulate shh components in the naïve state. A negative regulator of shh pathway (GPR161) is upregulated in the primed state, since its regulator microRNA is reduced. These two miRNA based control systems keep shh activity low in both states despite the emergence of cilia at the post-implantation stage.

## 2. Results

### 2.1. microRNAs Regulating Human Naïve to Primed ESCs Transition

To dissect how microRNAs effect pre-to-post-implantation transition in early development (Figure 1A) we identified microRNAs and their experimentally validated target genes showing consistent expression changes between naïve and primed pluripotency using published datasets (Table 1). We first identified 357 microRNAs that are significantly differentially expressed between human naïve and primed ESCs (Appendix A). We then identified 1146 protein-coding genes showing consistent differential expression in two naïve vs. primed studies [27,45]. We then used mirTarBase, a database of experimentally validated microRNA-target gene relations [46], to connect changes in microRNA and mRNA expression: a microRNA and its target gene is considered consistent if their expression levels change in the opposite direction in naïve vs. primed comparison. In total, we identified 2184 consistent microRNA-target gene connections between 280 out of the 357 differentially expressed microRNAs and 647 target genes out of the 1146 differentially expressed protein-coding genes (Appendix A). Consistent microRNA-target connections are cases where a microRNA is up (down)-regulated, and its target gene is down (up)-regulated (Figure 1A,B). Of the 647 target genes, 243 are significantly higher in naïve hESC compared to primed hESC; 404 target genes are significantly lower in naïve hESCs. Although the microRNA-target gene interactions have prior experimental support from mirTarBase, and show consistent expression changes in naïve vs. primed comparison, our analysis is still based on expression correlation. Therefore, the functional relevance of any specific microRNA-target gene interactions from our comprehensive list in naïve to primed transition still requires experimental validation. However, our integrative analysis does provide a high confidence starting point for future studies.

To identify biological processes under microRNA regulation in pre-to-post-implantation development, we performed Gene Ontology enrichment for the up- and down-regulated microRNA target genes. In this analysis, we compared significantly down-regulated microRNA-target genes against all significantly down-regulated genes (as opposed to all genes, which is typically done). The purpose is to identify whether microRNAs regulate specific biological processes. Target genes up-regulated in naïve hESCs (i.e., down-regulated in primed hESCs) did not show significant enrichment in specific pathways. Interestingly, we found that target genes down-regulated in naïve hESCs compared to primed hESCs (the corresponding microRNAs up-regulated in naïve hESC) are significantly enriched for mRNA transcription and multiple developmental processes (e.g., NOTCH3, ETV5, GLI2, PTCH1; Figure 1C). In particular, GLI2, a key transcription factor in the sonic hedgehog (shh) pathway, is significantly down-regulated in naïve hESC compared to primed hESC, and its microRNA regulators, miR-654-5p, miR-92a-2-5p, and miR-541-3p are up-regulated in naïve hESC compared to primed hESC.

### 2.2. Hh Pathway in Naïve-to-Primed Transition

Interestingly, multiple components of the shh pathway are downregulated in naïve hESC compared to primed hESC, including GLI1, GLI3, PTCH1, PITCH2 and SMO (Figure 2B). The shh pathway is an important regulator of embryonic development (Figure 2A, [49]). However, despite the presence of functioning components in the primed state, the shh pathway seems to play only a minimal role in the maintenance of pluripotency [50]. Binding of Nanog to shh components has been proposed as one of the mechanisms to inhibit shh pathway in pluripotent cells [51]. Our data suggest that microRNAs could down-regulate the expression of the components of shh-signaling pathway in naïve hESC (Figure 2H). Furthermore, naïve, preimplantation mouse ESC do not contain cilia, the key mammalian cellular organelle for Hh signaling. However, this organelle is found in the post-implantation stage [52,53,54]. To test the presence of cilia in human naïve-to-primed hESC transition we stained naïve and primed hESC with ARL13B. Importantly, we observed a significant increase in cilia number between these two stages (20-fold increase between naïve and primed hESC; (Figure 2C,D). These data argue that cilia morphogenesis occurs in naïve-to-primed hESC transition, licensing primed hESC for Hh pathway activity. It is therefore perhaps even more surprising that the Hh pathway is not active in the primed hESC stage. Through a whole genome CRISPR screen we found that GPR161, a ciliary G-protein coupled receptor, is required for the naïve to primed transition in hESC [55]. GPR161 is up-regulated in primed hESC (Figure 2B) and some of the microRNA predicted to target GPR161 (miR-372, miR-520, and miR-22) are down-regulated in primed compared to naive hESC [22,27]. GPR161 can negatively regulate the shh pathway [56], hence it could explain why the Hh-pathway is not active in primed hESC despite the presence of cilia and up-regulation of Hh-pathway components (Figure 2H). To test this hypothesis, we generated a GPR161 mutant hESC line using CRISPR Cas9 (Figure 2E–G). Most of the cells from the GPR161 mutant line express a GPR161 protein with a truncated C terminus due to the insertion of one nucleotide, resulting in a frameshift at residue 392 and the introduction of a premature STOP codon in position 426 (Figure 2E). The missing region in the truncated mutant line has been shown to be essential for GPR161 function and regulation of PKA and shh [57,58]. The mutant GPR161 hESC line exhibits primary cilia (Figure 2F) and one of the shh components, SMO, is up-regulated in GPR161 mutant hESCs compared to controls (Figure 2G). These data suggest that lack of GPR161 may induce promiscuous Hh pathway activity in hESCs. We propose that GPR161 is required for repressing the Hh pathway in the primed hESC stage (Figure 2H).

### 2.3. Metabolism in Naïve-to-Primed Transition

Naïve and primed hESCs have distinct metabolic profiles (Table 2 and Table 3, [26]). Some of these metabolic differences have been shown to be regulated by microRNAs [27,29]. We previously found that naïve hESCs can utilize fatty acids as an energy source, while primed hESCs cannot [27] (Table 2 and Table 3). Our analysis confirmed higher expression of miR-10a in naïve compared to primed hESC. Accordingly, miR-10a target genes in fatty acid activation (ACSL1), synthesis (FASN), elongation (ELOVL7), and desaturation (SCD) are downregulated in naïve compared to primed hESCs, suggesting that fatty acid synthesis is repressed in naïve hESC by critical miRNAs (Table 3). In contrast, CPT1A, a rate limiting transporter in fatty acid beta-oxidation is highly upregulated in naïve compared to primed hESC [27]. We found 11 microRNAs targeting CPT1A were significantly downregulated in naïve compared to primed hESC (miR-106b-5p, miR-20a-5p, miR-302c-3p, miR-17-5p, miR-20b-5p, miR-106a-5p, miR-302a-3p, miR-150-5p, miR-16-5p, miR-93-5p, miR-4517; see Appendix A for more details), suggesting that CPT1 and thereby beta-oxidation is down-regulated in primed hESC by critical miRNAs (Table 3). These data support our previous observation that fatty acid oxidation is more active in naïve hESCs compared to primed hESCs (Table 2 and Table 3). Thus, microRNAs may regulate the fatty acid oxidation (higher in naïve) vs. synthesis (higher in primed) differences between naive and primed hESCs (Table 3).

Additionally, naïve hESCs have higher oxidative phosphorylation (OXPHOS) activity than primed hESCs [27,59]. COX7B, a protein that is indispensable for cytochrome c oxidase (complex IV) assembly, activity, and mitochondrial respiration [60], is expressed higher in naïve than primed hESCs. Interestingly, hsa-miR-18a-5p and miR-17-5p that target COX7B are upregulated in in primed hESCs, and might therefore downregulate COX7B at that stage. miR-18a-5p and miR-17-5p belong to the miR-17~92 microRNA Cluster, which is a global regulator of cancer metabolism, known to suppress the expression of LKB1 [61]. Another member of the cluster, miR-20a, also targets CPT1A, a key step in fatty acid oxidation, suggesting the miR-17~92 cluster’s general role in the metabolic switch between naïve and primed hESC stages.

We further integrated microRNA expression, mRNA expression, and metabolomics data, and found consistent changes across three datasets in the polyamine metabolic pathway (Figure 3). ODC1 (ornithine decarboxylase) is a rate-limiting step in polyamine metabolism, and is significantly up-regulated in naïve hESCs. miR-218-5p, which targets ODC1, is down-regulated in naïve hESCs. Additionally, the substrate of ODC1, ornithine, is significantly depleted in naïve hESCs based on our metabolomics data. Spermidine, a key product of polyamine synthesis, is more abundant in naïve hESCs. Consistent with the conversion of ornithine to spermidine, the alternative route of ornithine catabolism, citrulline, is less abundant in naïve hESCs. Other enzymes or metabolites in the polyamine pathway did not change, possibly because some enzymes are regulated at the translational level, not transcriptional level (e.g., AMD1) [67]. ODC1 has been shown to be essential for mouse ESC self-renewal [68], and high polyamine (e.g., spermidine) levels have been shown to promote mouse ESC self-renewal [69]. Our analysis suggests that ODC1 and spermidine may also promote human naïve pluripotency, and down-regulation of miR-218-5p may be important for maintaining high polyamine metabolism in naïve hESCs. miR-218-5p has been widely studied in cancer [70].

### 2.4. microRNAs Regulating Mouse Naïve to Primed ESCs Transition

microRNAs also play important roles in regulating pluripotency in mouse. Several studies have profiled microRNAs in mouse embryonic stem cells and epiblast stem cells, or during mouse embryonic stem cell differentiation [38,39,40] (Table 1). To generate a robust consensus set of microRNAs associated with mouse naïve to primed transition, we integrated microRNA-seq datasets from three different studies. We identified 221 microRNAs that show consistent and significant changes in at least two out of the three studies (same direction and at least 1.5-fold change for either the -3p or -5p form, Appendix A and Methods). Similar to the human microRNA analysis, we combined these consistently changing microRNAs with mirTarBase information, and mESC vs. mEpiSC mRNA-seq data, and found 836 consistent microRNA-target gene interactions between 119 microRNAs and 304 target genes (Figure 4A, only a subset of the 221 microRNAs are connected to target genes changing in the opposite direction. The full list of 836 interactions are in Appendix A).

Through Gene Ontology enrichment analysis, we found that compared to all down-regulated genes, down-regulated genes that are also targets of these consistently changing microRNAs are significantly enriched for multiple developmental processes (Figure 4B). This is similar to human naïve to primed transition, where microRNAs also target developmental genes (Figure 1B).

MicroRNAs also regulate the metabolic differences between naïve and primed states in mouse, especially in amino acid metabolism based on our analysis. We identified two metabolic genes, Slc1a2 and Slc25a12, that satisfy the following three criteria: (1) significant transcriptomic change; (2) their microRNA regulators change in the opposite direction; and (3) their connected metabolite (e.g., substrate/product) show significant change between naïve and primed mESCs. Slc1a2, a glutamate transporter, is 11.7-fold higher in mEpiSC compared to mESC, while its microRNA regulator, miR-199b-3p, is consistently up-regulated in all naïve state (down-regulated in mEpiSC) in three microRNA-seq studies. Slc1a2 is important for glutamate homeostasis [71]. Consistently, glutamate level is 1.93-fold higher in mEpiSC compared to mESC based on our metabolomics data. Glutamate can be converted to α-ketoglutarate and utilized in the TCA cycle. Higher expression of the glutamate transporter and abundance of glutamate in EpiSC suggest that glutamate may be an important fuel for EpiSC. Supporting this hypothesis, uptake of glutamate and its conversion to α-ketoglutarate has been previously reported during the initial differentiation of human primed pluripotent stem cells [72].

Slc25a12 (mitochondrial aspartate/glutamate carrier, commonly called aralar) is 5.5-fold higher in mESC compared to EpiSC. Its microRNA regulators (miR-466h-5p, 466m-5p, and 669m-5p) are all lower in mESC. Slc25a12 is a critical component of the mitochondria malate-aspartate shuttle, and deficient Slc25a12 expression is known to increase lactate production from pyruvate, and reduce glucose oxidation in the brain [73]—a metabolic phenotype similar to the primed state. Additionally, deficient Slc25a12 expression results in reduced aspartate level in the brain [74], and our metabolomics data showed that aspartate is significantly lower in EpiSC (where Slc25a12 expression is lower) compared to mESC. The expression pattern of Slc25a12 and abundance of aspartate may influence the shift from glucose oxidation to glycolysis and lactate production in the naïve to primed transition.

### 2.5. Consistent microRNA Changes across Human and Mouse Pluripotency

Previous studies have shown that certain microRNAs such as let-7 and microRNAs in the imprinted Dlk1-Dio3 locus are important regulators of both human and mouse pluripotency [10,39]. To systematically identify more shared microRNAs regulating both human and mouse pluripotency, we focused on 115 microRNAs (Appendix A) that satisfy two criteria: (1) that show significant differences between human naïve and primed ESCs; and (2) that change in the same direction in at least one mouse naïve vs. primed microRNA-seq studies (>1.5-fold). We recovered members of the let-7 family (let7-b, d, f, and i) and microRNAs from the Dlk1-Dio3 locus (miR-541-5p, miR-410-3p, miR-381-3p, and miR-495-3p) as consistently higher in human and mouse naïve ESCs. We also found miR-143-3p to be the only microRNA that is 1.5-fold higher in naïve ESCs across all four different datasets. miR-143 is significantly induced by vitamin C and vitamin C is known to promote naïve pluripotency [75]. Additionally, miR-143 over-expression reduced differentiation and up-regulates key pluripotency genes (Oct4, Klf4, and Esrrb) [76]. Thus, this list of 115 consistently changing microRNAs provide a useful resource to mine microRNAs regulating pluripotency state transitions. We further identified microRNAs showing consistent changes in three or more pluripotency studies (Table 4).

To reveal the potential functions of these 115 microRNAs, we used Co-expression Meta-analysis of miRNA Targets (CoMeTa) [77] to assign enriched Gene Ontology terms to 30 of the 115 microRNAs (Appendix A). These inferred functions include small GTPase signaling (miR-582, higher in naïve), angiogenesis (miR-369, higher in naïve), cell death (miR-338-3p, higher in primed), and cell adhesion (miR-671-5p, higher in primed). Previous studies have found that low cell-matrix adhesion promotes homogeneous self-renewal of naïve mESCs [78], which supports the computationally predicted function of miR-671-5p.

Of the 115 microRNAs, 73 have at least one target gene that changes in the same direction between human and mouse. That is, both the microRNA and their target genes change in the same direction in both species (Appendix A). Some of the consistently changing microRNAs regulate the metabolic transitions between naïve and primed pluripotency. For example, miR-615-3p is higher in the naïve state, and its target gene ATP13A2 is lower in both human and mouse naïve stem cells. ATP13A2 is important for mitochondria maintenance, and ATP13A2 knockdown cells show increased oxygen consumption [79]. The increased oxygen consumption in ATP13A2 knockdown cells is similar to naïve stem cells where ATP13A2 expression is lower, and oxygen consumption is higher compared to primed stem cells [27]. Similarly, miR-485-5p is higher, while its target gene HIF3A is lower in both human and mouse naïve state. HIF3A expression is known to increase in response to hypoxia in pluripotent stem cells [80], and its lower expression in the naïve state agrees with our previous results that the HIF pathway is activated in the primed state but not in naïve [27]. Additionally, VLDLR (very low density lipoprotein receptor) is higher in primed (lower in naïve), and three microRNAs targeting VLDLR, miR-493-3p, 376c-3p, and 409-5p are all lower in primed (higher in naïve). Higher VLDLR expression is associated with increased amount of lipid droplet formation, which is consistent with our previous observation [27]. Six microRNAs (miR-302c-3p, miR-20b-5p, miR-106a-5p, miR-302a-3p, miR-455-3p, and miR-338-3p) targeting ACOT9 (Acyl-CoA thioesterase 9) are all lower in naïve; and ACOT9 is higher in naïve. ACOT9 hydrolyzes both short- and long-chain acyl-CoAs, and links amino acid and fatty acid metabolism in the mitochondria [81].

### 2.6. microRNAs and Their Target Genes Involved in Diapause Regulation

The transition from pre-to-post-implantation can be suspended in an intermediate state called diapause. To obtain a more complete picture of microRNAs potentially involved in regulating the diapause state, we combined a publicly available microarray-based profiling of diapause and re-activated embryos [37] with our RNA-seq of diapause and post-implantation embryos [35], using the strategy outlined in Figure 1A. We identified 379 consistent connections between 38 microRNA and 274 target genes, where a microRNA is up (down)-regulated, and its target gene is down (up)-regulated (Appendix A). We compared target genes up-regulated in diapause (their microRNA regulators down in diapause) against all up-regulated genes in diapause, and found that these microRNA target genes show more enrichment in stress response (e.g., EGR1) [82], hypoxia (BACH1) [83], and hormone response (e.g., ZBTB7A) [84] pathways, which is consistent with the diapause phenotype (Figure 5A). On the other hand, target genes down-regulated in diapause (their microRNAs up-regulated in diapause) showed enrichment in cellular morphogenesis, differentiation, and development (Figure 5B), which is consistent with diapause as a suspended state in developmental progression.

## 3. Materials and Methods

Raw mRNA-seq data from Sperber et al. [27] and Theunissen et al. [45] were aligned to hg19/GRCh37 with STAR aligner57. Transcript quantification was performed with htseq-count from HTSeq package58 using GENCODE v15. Differential expression analysis was performed with DESeq after filtering out genes whose total read count across samples are below the 40th quantile of all genes. RNA-seq of mouse ESC and EpiSC from Factor et al. [48] was analyzed similarly using the mouse genome reference. Differentially expressed microRNAs were downloaded from Appendix A from Sperber et al. [27], Jouneau et al. [38], Gu et al. [40], and Moradi et al. [39]. topGO R package [85] was used for Gene Ontology enrichment analysis. Processed metabolomics data were downloaded from Sperber et al. [27]. Single cell RNA-seq of monkey in vivo implantation (Figure 3C) was downloaded from Nakamura et al. [86]. We noticed that when previous studies established the role of microRNAs in pluripotency (e.g., miR-290), they did not distinguish miR-290-3p vs. miR-290-5p (Gu et al., Cell Research 26:350–366 (2016)). We therefore combined the -3p/-5p form when analyzing the three mouse microRNA-seq datasets. A microRNA is considered consistently changing if any one of the three criteria is satisfied: (1) the -3p form changed ≥1.5-fold in at least two out of three studies; (2) the -5p form changed ≥1.5-fold in at least two out of three studies; (3) the -3p form changed 1.5-fold in one study, the -5p form changed >1.5-fold in the same direction as the -3p form, but in a different study.

### 3.1. Culture of Naïve and Primed Human Pluripotent Stem Cells

Naïve hESC Elf-1 (NIH_hESC Registry #0156, [42]) and primed human iPSC (WTC-11) [87] were cultured as previously described [26,27,43,44]. Naïve cells were grown on a feeder layer of irradiated primary mouse embryonic fibroblasts (MEF) in naïve hESC 2iL-I-F media: DMEM/F-12 media supplemented with 20% knock-out serum replacer (KSR), 0.1 mM nonessential amino acids (NEAA), 1 mM sodium pyruvate, penicillin/streptomycin (all from Invitrogen, Carlsbad, CA, USA), 0.1 mM β-mercaptoethanol (Sigma-Aldrich, St. Louis, MO, USA), 1 µM GSK3 inhibitor (CHIR99021, Selleckchem, Houston, TX, USA), 1 µM of MEK inhibitor (PD0325901, Selleckchem), 10 ng/mL human LIF (Chemicon, Temecula, CA, USA), 5 ng/mL IGF1 (Peprotech, Rocky Hill, NJ, USA) and 10 ng/mL bFGF. Cells were passaged using 0.05% Trypsin-EDTA (Life Technologies, Carlsbad, CA, USA). Cells were transferred to matrigel-coated plates prior molecular analysis. Naive Elf1 cells were pushed to the primed state by culturing the cells in mTeSR1 media (StemCell Technologies, Vancouver, BC, Canada) on matrigel-coated plates. Primed pluripotent stem cells WTC11 were cultured in mTeSR1 media on Matrigel-coated plates. All cells were cultured at 37 °C, 5% CO_2_, and 5% O_2_.

### 3.2. Generation of GPR161 Mutant hESC

Naïve hESC (Elf1 2iL-I-F) were spin-infected with lentiCRISPR-v2 lentiviral prep expressing GPR161 sgRNA (CCCACACCTCACTGCGCTCA) in presence of 4 μg/mL polybrene and plated onto irradiated DR4 MEF. Two days later, cells were selected with puromycin (0.5 µg/mL, for 3 days). Genomic DNA was extracted using DNAzol reagent (Invitrogen) according to manufacturer’s instructions and quantified using Nanodrop ND-1000. Genomic regions flanking the CRISPR target sites were PCR amplified (primers Fwd: TCACCTTGGTGGTGGTTGAC; Rev: AAAACGCGACAGGTGAGAGG) and sent for Sanger sequencing. Analysis of the CRISPR editing data through the ICE tool software (ICE v2; Synthego, Sunnyvale, CA, USA) predicted a score of 73% KO in the pooled population.

### 3.3. Immunofluorescence Staining and Confocal Imaging

Naïve and primed pluripotent stem cells were cultured on Matrigel-coated glass cover slips at the bottom of the culture dishes. Cells are fixed with 4% paraformaldehyde (PFA) for 15 min, washed with PBS 1X (3 times, 5 min each) and blocked with 3% BSA (VWR 0332) and 0.1% Triton (Sigma 9002-93-1) (blocking solution) for 1 h at room temperature (RT). Cells were then incubated overnight with ARL13B- (Proteintech 17711-1-AP, 1:250), and ZO-1 (Invitrogen 33-9100, 1:250) antibodies diluted in blocking solution. After PBS 1X washes (3 times, 5 min each) cells were incubated with anti-rabbit Alexa 488 and anti-mouse 647 conjugated secondary antibodies (1:250) at RT for 1 h. The slides were then washed 4 times with PBS 1X and stained with DAPI DAPI during the 2nd wash (1:10) for 10 min. Glass cover slips were mounted on slides with vectashield Hardset (H-1400). The images were taken on Leica SPE confocal and Nikon A1 confocal and analyzed using Fiji ImageJ, v1.52g. The percentage of cells with the presence of a cilia was assessed by counting Arl-13B positive cells. At least 500 cells were counted for each condition.

### 3.4. RT-qPCR Analysis of GPR161 Mutant

RNA was extracted using Trizol (Life Technologies) according to manufacturer’s instructions. RNA samples were treated with Turbo DNase (ThermoFischer, Waltham, MA, USA) and quantified using Nanodrop ND-1000 (Thermo Scientific). Reverse transcription was performed using iScript reverse transcription kit (BioRAD, Hercules, CA, USA). 10 ng of cDNA was used to perform qRT-PCR using SYBR Green (Applied Biosystems, Foster City, CA, USA). Real-time RT-PCR analysis was performed on 7300 real time PCR system (Applied Biosystems) and β-actin was used as an endogenous control. Primer sequences for SMO are Fwd: 5′-ACGAGGACGTGGAGGGCTG-3′ and Rev: 5′-CGCACGGTATCGGTAGTTCT-3′; and for β-actin Fwd: TCCCTGGAGAAGAGCTACG and Rev: GTAGTTTCGTGGATGCCACA.

## 4. Conclusions

microRNAs play important roles in regulating cell fate choices. Through integrative analysis of mRNA-seq, microRNA-seq, and metabolomics data in human and mouse, we identified a comprehensive resource for microRNAs and their target genes involved in regulating the developmental and metabolic processes in naïve to primed transition, as well as in their paused, intermediate stage, diapause.

We identified a set of microRNAs with robust changes across multiple studies in both human and mouse by integrating transcriptomic data of mRNA and microRNA, as well as metabolomics data of naïve to primed transition in both human and mouse. The target genes of these consistently changing microRNAs regulate important metabolic and developmental pathways in naïve to primed transition. Importantly, we propose that the dynamics of the paused stage, embryonic diapause are affected by microRNA regulated stress and inflammation response genes. This is interesting in the light of recent publications that have revealed importance of stress response and inflammation in regulation of embryonic diapause and stem cell quiescence [35,88,89], and calls for further analysis on microRNA contribution in stress induced regulation of pause in development.

These studies also propose that microRNAs could play an important role in the maintenance of pluripotency. In particular, we suggest that miRNAs repress Hh pathway activity in both naïve and primed hESCs, preventing their differentiation. Interestingly we found in human ESC, as previously observed in mouse embryos, that primary cilia appear only at the primed stage of hESCs, in naïve hESCs the presence of primary cilia is minimal. Since primary cilium is essential for Hh-pathway activity, this could explain the lack of Hh signaling in the naïve pluripotent stage. Furthermore, microRNA may target components of the shh pathway at the naïve hESC stage. Our data suggest that microRNAs could target a repressor of shh, the G-protein coupled receptor GPR161, usually located in primary cilia [56]. As cells transition from the naïve to primed stage, primary cilia are formed and Hh-pathway components, for example GPR 161, PTCH, GLI, and SMO are expressed, however, the Hh-pathway is not activated. We now propose that in order to keep the Hh-pathway inactive in the primed hESC, the cells utilize GPR161. In the naïve to primed transition microRNAs targeting GPR161 are downregulated and GPR161 is expressed, resulting in the repression of the Hh pathway activity, despite the existing cilia and expression of the Hh pathway components. In summary, our study provides a valuable resource for dissecting microRNA based regulation of naïve to primed transition.

## Figures and Tables

**Figure 1 ijms-20-05864-f001:**
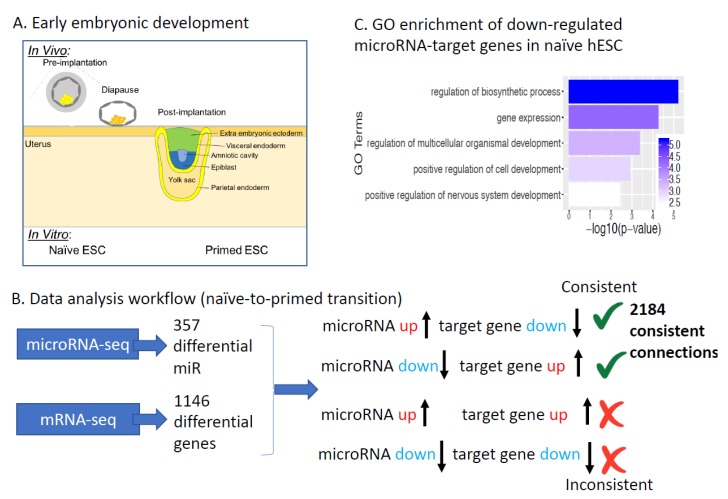
microRNAs regulating human naïve to primed ESCs transition: (**A**) A schematic figure of early embryonic developmental stages. (**B**) Analysis workflow. We first identified 357 differentially expressed microRNAs and 1146 differentially expressed protein-coding genes in two naïve-primed studies [27,45]. We then used mirTarBase to connect changes in microRNA and their experimentally validated target genes, and filtered down to 2184 miR-target gene connections where microRNA is up and its target is down (or vice versa). Green √ means the microRNA-gene connection is considered consistent; red × means the connection is not consistent. (**C**) Gene ontology enrichment of microRNA target genes with lower expression in human naïve ESCs (the microRNA regulators are higher in naïve). x-axis is negative log10 of enrichment *p*-value (larger means more significant).

**Figure 2 ijms-20-05864-f002:**
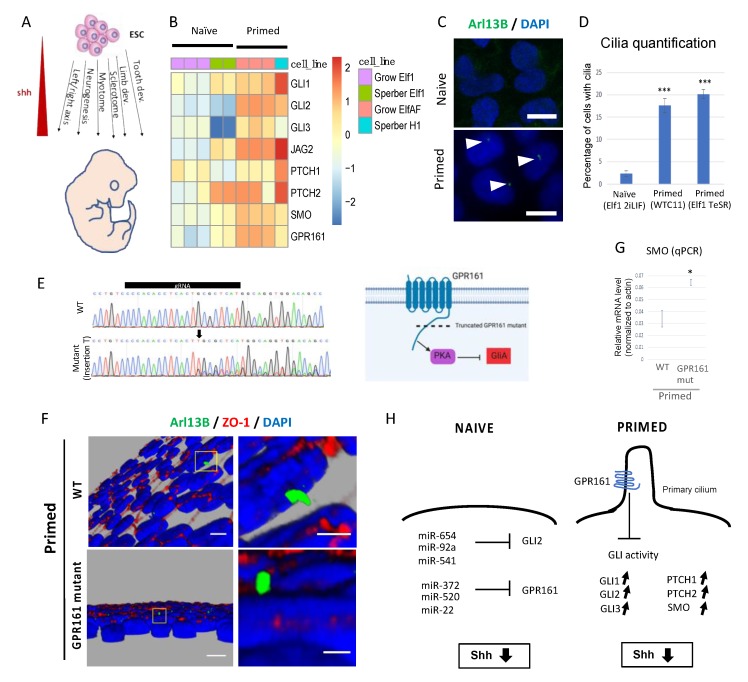
The Hh pathway in naïve-to-primed transition: (**A**) Schematic representation of the role of shh pathway during early development. Shh pathway is not active in ESC but plays an important role in various developmental processes. (**B**) Expression of shh pathway components in naïve hESC (Elf1 2iLIF, Sperber et al. and Grow et al.) and primed hESC (Elf1 AF, Grow et al. and H1 Sperber et al.). (**C**,**D**) Cilia are present only in primed hESC, not in naïve hESC, as visualized by Arl13B staining in naïve (Elf1 2iLIF) and primed (Elf1 TeSR) hESC. Scale bars represent 5 μm, triangles indicate the presence of primary cilia. Standard Error of Mean (s.e.m); *** *p* < 0.001; 2-tailed *t*-test. (**E**) Sanger sequencing results of the GPR161 gene around the gRNA in Elf1 wild type (WT) and GPR161 mutant show an insertion of the nucleotide T, resulting in a frame shift and premature STOP codon (**left panel**). This mutation leads to the generation of a truncated GPR161 protein missing the functional C terminus domain responsible for regulation of PKA and shh activity (**right panel**). (**F**) Cilia (ARL13B) immunofluorescence staining in Elf1 WT and Elf1 GPR161 mutant grown in primed conditions (TeSR). On the right side are high magnification images of the boxed regions shown on the left side. Scale bars represent 5 μm. (**G**) Shh component SMO is up-regulated in Elf1 GPR161 mutant cells compared to WT in primed conditions (qPCR analysis). S.e.m.; * *p* < 0.05; 2-tailed *t*-test. (**H**) Model of repression of shh activity in naïve and primed hESC through microRNAs. T-bars indicate repression, arrow upwards upregulation and downwards downregulation.

**Figure 3 ijms-20-05864-f003:**
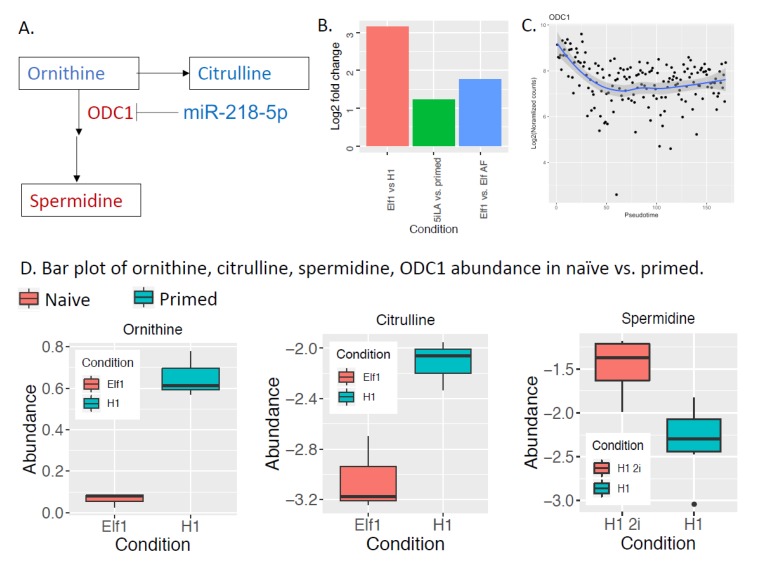
The polyamine pathway is dysregulated in naïve to primed transition: (**A**) Ornithine is converted to polyamines (spermine and spermidine) by the rate limiting enzyme ornithine decarboxylase (ODC1). ODC1 is consistently up-regulated in the naïve state, and its microRNA regulator, miR-218-bp, is lower in the naïve state. Each rectangle box is a metabolite; each arrow is a reaction. Red means higher abundance in naïve state; blue means lower abundance in naïve state. “--|” (T-bar) indicates repression of ODC1 by miR-218-5p. (**B**) Log2 fold change of ODC1 in three separate naïve vs. primed cell line comparisons. (**C**) ODC1 expression changes in monkey pre-to-post-implantation transition in vivo. ODC1 is also expressed higher in the naïve state *in vivo*. Blue line represents a local regression fit of the data (**D**) Bar plot of ornithine, citrulline, and spermidine abundance in naïve (Elf1, H12i) and primed (H1) cell lines.

**Figure 4 ijms-20-05864-f004:**
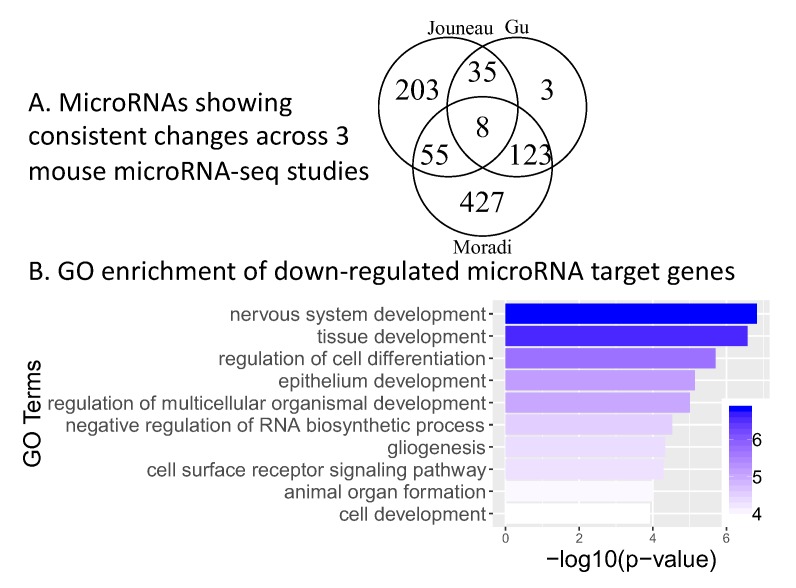
microRNAs regulating mouse naïve to primed ESCs transition: (**A**) Venn diagram of microRNAs changing in the same direction in naïve to primed transition across three microRNA-seq studies of mouse ESC vs. mouse EpiSC. (**B**) Gene ontology enrichment of microRNA target genes with lower expression in mouse ESCs (the microRNA regulators are higher in naïve). The x-axis is the negative log10 of enrichment *p*-value (larger means more significant).

**Figure 5 ijms-20-05864-f005:**
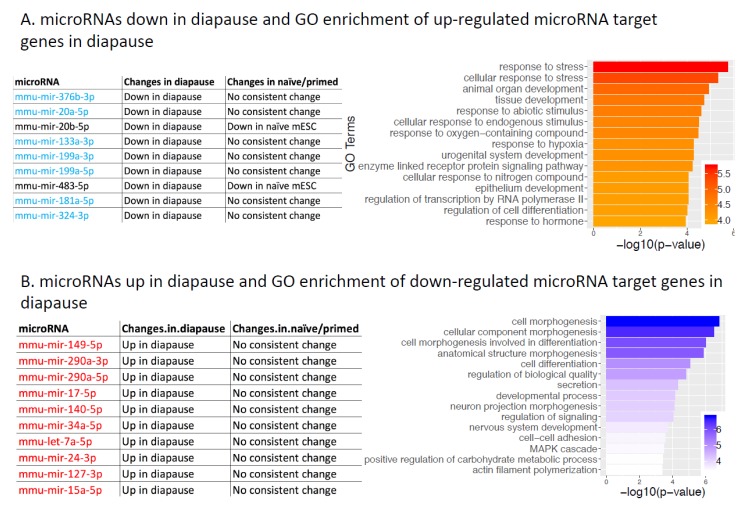
Consistent microRNA changes across human and mouse pluripotency: (**A**) **Left panel**: microRNAs with lower expression in the diapause state compared to reactivated state. MicroRNAs that are expressed lower only in diapause but not in the naïve state are highlighted in blue. **Right panel**: Gene ontology enrichment of microRNA target genes with higher expression in mouse diapause embryos compared to post-implantation embryos (the microRNA regulators are lower in diapause, shown on the **left panel**). The x-axis is the negative log10 of enrichment *p*-value (larger means more significant). (**B**) **Left panel**: Top 10 microRNAs with highest expression in the diapause state compared to reactivated state. These microRNAs are highlighted red because they are higher in diapause but not in naïve state. **Right panel**: Gene ontology enrichment of microRNA target genes with lower expression in mouse diapause embryos compared to post-implantation embryos (the microRNA regulators are higher in diapause, shown on the left panel).

**Table 1 ijms-20-05864-t001:** Datasets used in this study.

Study	Type	Condition	Growth Media	Accession Number
Sperber H. et al. [27]	microRNA-seq	Naïve hESC	2iL-I-F	GSE60995
Sperber H. et al. [27]	mRNA-seq	Naïve and primed hESC	2iL-I-F/KSR + FGF	GSE60995
Jouneau A. et al. [38]	microRNA-seq	Mouse ESC and EpiSC	15% FBS-LIF/AF	Supplemental data from original paper
Gu et al. [40]	microRNA-seq	Mouse ESC differentiation time course	N2B27-2iLIF/KSR + FGF	Supplemental data from original paper
Moradi S. et al. [39]	microRNA-seq	Mouse ground ESC and naïve ESC	N2B27 2iLIF or 3iLIF/15% FBS-LIF	GSE87174
ENCODE project	microRNA-seq	primed ESC	TeSR	https://www.encodeproject.org
Roadmap Epigenome project	microRNA-seq	primed ESC	TeSR	http://www.roadmapepigenomics.org
Theunissen et al. [45]	mRNA-seq	Naïve/primed hESC	5iLA/serum	GSE59435
Grow et al. [47]	mRNA-seq	Naïve/primed hESC	2iLIF/KSR + FGF	GSE63570
Factor et al. [48]	mRNA-seq	Mouse ESC and EpiSC	KSR LIF/KSR + FGF	GSE57409
Hussein et al. [35]	mRNA-seq	Mouse pre-implantation, diapause, post-implantation	From embryos	
Liu et al. [37]	microRNA microarray	Mouse diapause and re-activated embryos	From embryos	Supplemental data from original paper

**Table 2 ijms-20-05864-t002:** Metabolic differences between naïve and primed pluripotent stem cells.

Naive	Primed	Assay	References
Glycolysis ↑	Glycolysis ↑	RNAseq, Metabolomics, Seahorse Flux Analyzer, TMRE	Zhou et al. [62] Takashima et al. [59] Sperber et al. [27] Gu et al. [28] Zhang et al. [63] Sun et al. [64] Baha et al. [65] Cornacchia et al. [66]
OXPHOS ↑	OXPHOS ↓		
Fatty acid synthesis ↓	Fatty acid synthesis ↑	RNAseq, Metabolomics, Seahorse Flux Analyzer, Oil RedO and Bodipy staining	Sperber et al. [27] Cornacchia et al. [66]
Fatty acid oxidation ↑	Fatty acid oxidation ↓		

↑ indicates up-regulation, ↓ indicates down-regulation.

**Table 3 ijms-20-05864-t003:** Hypothetic role of microRNAs in regulation of fatty acid metabolism of naïve and primed embryonic stem cells (ESC).

Naive	Primed	Assay	References
FA synthesis, activation, elongation, desaturation(FASN, ACSL1, ELOVL7, SCD) 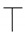 miR-10a	FA transporter (rate limiting FAO)CPT1A 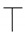 miR-106b-5p, miR-20a-5p, miR-302c-3p, miR-17-5p, miR-20b-5p, miR-106a-5p, miR-302a-3p, miR-150-5p, miR-16-5p, miR-93-5p, miR-4517	RNAseq, microRNAseq, RT-qPCR	Sperber et al. [27]

T bars indicate microRNA based repression of the target genes.

**Table 4 ijms-20-05864-t004:** microRNAs showing consistent changes in at least three out of four datasets.

ID	log2FC.Ref_Epigenome	log2FC.ENCODE	padj.Ref_Epigenome	padj.ENCODE	Jouneau	Gu	Moradi	How Many Consistent
hsa-mir-143, hsa-miR-143-3p	4.10	7.50	2.932 × 10^−3^	1.163 × 10^−7^	2.16	1.70	0.79	4
hsa-mir-200c, hsa-miR-200c-3p	−6.60	−8.97	8.956 × 10^−96^	5.294 × 10^−93^	6.44	−1.03	−0.88	3
hsa-mir-205, hsa-miR-205-5p	3.23	9.28	4.715 × 10^−12^	1.392 × 10^−88^	2.23	0.93	−1.28	3
hsa-mir-302c, hsa-miR-302c-3p	−1.68	−6.40	5.552 × 10^−13^	1.086 × 10^−66^	−5.47	−1.12	2.63	3
hsa-mir-20b, hsa-miR-20b-5p	−1.93	−5.37	1.923 × 10^−16^	2.778 × 10^−51^	0.00	−1.60	−0.75	3
hsa-mir-335,hsa-miR-335-5p	−1.76	−5.07	1.075 × 10^−13^	6.196 × 10^−47^	0.00	−1.00	−0.74	3
hsa-mir-302a, hsa-miR-302a-3p	−1.50	−4.46	1.090 × 10^−10^	8.146 × 10^−39^	−5.15	−2.08	1.67	3
hsa-mir-412, hsa-miR-412-5p	7.17	15.02	3.547 × 10^−64^	1.038 × 10^−28^	−2.60	0.92	3.34	3
hsa-mir-433, hsa-miR-433-3p	5.38	15.02	6.536 × 10^−39^	7.545 × 10^−21^	0.00	0.62	4.29	3
hsa-mir-495, hsa-miR-495-3p	4.68	7.47	2.492 × 10^−34^	1.098 × 10^−19^	0.00	0.60	4.27	3
hsa-mir-582, hsa-miR-582-3p	2.72	5.91	7.063 × 10^−12^	4.664 × 10^−14^	1.40	0.96	0.48	3
hsa-mir-196a-1, hsa-miR-196a-5p	8.96	15.02	1.038 × 10^−15^	1.389 × 10^−13^	0.00	2.46	0.78	3
hsa-mir-196a-2, hsa-miR-196a-5p	8.69	15.02	2.845 × 10^−12^	2.660 × 10^−12^	0.00	2.46	0.78	3
hsa-mir-200a, hsa-miR-200a-5p	−3.37	−4.71	1.511 × 10^−6^	5.775 × 10^−9^	0.00	−0.89	−1.59	3
hsa-mir-1247, hsa-miR-1247-3p	3.40	15.02	1.677 × 10^−8^	4.527 × 10^−6^	2.85	0.00	2.17	3
hsa-mir-708, hsa-miR-708-3p	1.13	1.45	5.997 × 10^−5^	3.027 × 10^−4^	1.87	0.72	0.08	3
hsa-mir-199b, hsa-miR-199b-3p	3.34	4.27	4.641 × 10^−2^	6.808 × 10^−3^	2.54	0.30	1.01	3
hsa-let-7f-2, hsa-let-7f-5p	4.79	4.43	1.363 × 10^−2^	8.710 × 10^−3^	0.00	1.97	0.78	3
hsa-let-7f-1, hsa-let-7f-5p	4.72	4.31	1.455 × 10^−2^	9.959 × 10^−3^	0.00	1.97	0.78	3
hsa-mir-182, hsa-miR-182-3p	3.19	4.10	7.430 × 10^−4^	2.571 × 10^−2^	0.00	0.64	0.88	3

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
