# Peer review of "microRNAs Regulating Human and Mouse Naïve Pluripotency"

_ijms, 2019, doi:10.3390/ijms20235864_

Round 1

Reviewer 1 Report

In the paper entitled ‘'microRNAs regulating human and mouse naïve pluripotency,'' the authors have combined multiple published databases of microRNA-seq, RNA-seq and metabolomics datasets from human and mouse in order to characterize changes that are taking place from naïve to primed transition. From this analysis, they show that the formation of primary cilium and remodeling of metabolic processes are essential events targeted and controlled by microRNA sets differentially regulated during the transition. Moreover, they have identified 115 microRNAs that significantly change in the same direction in naïve to primed transition in both human and mouse, many of which are novel candidate regulators of pluripotency. Furthermore, we identified 38 microRNAs and 274 target genes that may be involved in diapause.

This paper is indeed a comprehensive descriptive study of the microRNAs that mark specific stages of pluripotency. This study is presenting the correlation of events which does not imply causation unless the authors decide to conduct mechanistic studies.

In that sense when they describe Fig.3, line 191, and they mention that ‘' We further integrated microRNA expression, mRNA expression and metabolomics data found consistent across three datasets in the polyamine metabolic pathway (Fig. 3)'' there are a couple of points that need to be clarified so as not to oversimplify. The authors checked three components of the polyamine metabolic pathway, and they do mention the reasoning why they picked these three components of the pathway, but they should also present what happens to other components of that pathway.

Similarly, in lines 228 and 238, they chose to present the correlations of Slc1a2 and Slc25A12, respectively, and it is not clear why they chose to discuss these two examples specifically. They need either to clarify out of how many candidates they picked these two examples, or if these two examples were the only candidates that came up. The importance here is not to bias the reader with their perspective.

I think this study should be published because it provides a very comprehensive descriptive study of the miRNAs that mark different stages of pluripotency, but it needs to be clarified in the discussion that the study doesn't identify targets, because without experimental validation no conclusions can be drawn, but it suggests potential targets to be studied in the future. Also, it needs to be emphasized that this is a presentation of correlations.

Author Response

In the paper entitled ‘'microRNAs regulating human and mouse naïve pluripotency,'' the authors have combined multiple published databases of microRNA-seq, RNA-seq and metabolomics datasets from human and mouse in order to characterize changes that are taking place from naïve to primed transition. From this analysis, they show that the formation of primary cilium and remodeling of metabolic processes are essential events targeted and controlled by microRNA sets differentially regulated during the transition. Moreover, they have identified 115 microRNAs that significantly change in the same direction in naïve to primed transition in both human and mouse, many of which are novel candidate regulators of pluripotency. Furthermore, we identified 38 microRNAs and 274 target genes that may be involved in diapause.

This paper is indeed a comprehensive descriptive study of the microRNAs that mark specific stages of pluripotency. This study is presenting the correlation of events which does not imply causation unless the authors decide to conduct mechanistic studies.

In that sense when they describe Fig.3, line 191, and they mention that ‘' We further integrated microRNA expression, mRNA expression and metabolomics data found consistent across three datasets in the polyamine metabolic pathway (Fig. 3)'' there are a couple of points that need to be clarified so as not to oversimplify. The authors checked three components of the polyamine metabolic pathway, and they do mention the reasoning why they picked these three components of the pathway, but they should also present what happens to other components of that pathway.

Response: Other components of the polyamine pathway did not change significantly between naïve and primed ESCs based on our transcriptional or metabolomic data. The three components that did change (the metabolites ornithine and spermidine; the enzyme ODC1) have significant roles. Ornithine is the starting point of the polyamine pathway, and ODC1 is the rate-limiting step in the pathway. Other components, such as AMD1 (adenosyl methionine decarboxylase) is known to be regulated at the translational level.

Similarly, in lines 228 and 238, they chose to present the correlations of Slc1a2 and Slc25A12, respectively, and it is not clear why they chose to discuss these two examples specifically. They need either to clarify out of how many candidates they picked these two examples, or if these two examples were the only candidates that came up. The importance here is not to bias the reader with their perspective.

Response: We clarified in the text that these two genes are picked because the corresponding metabolites that they transport also showed significant changes in naïve to primed transition. These two genes are chosen out of 40 metabolic genes that show anti-correlation with the 221 consistently changing microRNAs in mouse.

I think this study should be published because it provides a very comprehensive descriptive study of the miRNAs that mark different stages of pluripotency, but it needs to be clarified in the discussion that the study doesn't identify targets, because without experimental validation no conclusions can be drawn, but it suggests potential targets to be studied in the future. Also, it needs to be emphasized that this is a presentation of correlations.

Response: The microRNA-target gene relations we presented satisfy two criteria: first, they are included in the mirTarBase, a database of experimentally validated microRNA-target gene relations. Second, the microRNA and target genes exhibit opposite expression pattern in the naïve vs. primed comparison. Therefore, these relations have additional evidence besides expression correlation. However, we do agree that without experimental validation, we do not know if the specific microRNA-target gene relations are important for naïve to primed transition. We followed the reviewer’s suggestion and clarified that the consistent microRNA-target gene relations is largely based on expression correlation and need to be functionally validated.

Reviewer 2 Report

Title:    MicroRNAs regulating human and mouse naïve pluripotency

The article concerns the topic of microRNAs which regulate human and mouse naïve to primed transition of pluripotent stem cells as well as diapause stage of this transition. This paper is a meta-analysis of micro-RNA-seq, RNA-seq and metabolomics datasets form human and mouse pluripotent cells. In my opinion the results presented in this article are not novel, however it is a good database for scientist working with pluripotent stem cells.

As I understand, the meta-analysis does not need additional results to confirm the high throughput data. However, if authors decided to add some of their own results in Fig. 1 thus I must suggest to add also other crucial results to confirm the stated data: activation of Hedgehog signaling pathway (maybe on/off mechanism), functional assessment of metabolism (LDH activity, OXPHOS activity, glutamate assay).

In my opinion, the manuscript is clearly written and easy to understand. However I found some mistakes and phrases which need explanation:

Line 20 - Hh-pathway – explain the abbreviation

Line 31 - Start the sentence with a capital letter (please correct throughout the manuscript)

Line 31 (and throughout the manuscript) – Please add the space before the citation and delete after

Line 65 – it should not be written in italics

Fig. 2C and F – The figures are too small

Fig. 2G – Description is for fig. G, however on the figure there is no “G” but “H”

Fig. 4 - Figure is too big comparing to others. The description: Figure 4, should be deleted from above the Figure.

Methodology section – the more  detailed description of high throughput data should be added. Also the GSE numbers from GEO database of all presented results should be added.

The text is well written, presented figures fulfills the raised topic. They are also well described and discussed. The reference are well cited. In my opinion manuscript should be accepted for publication after minor revision.

Author Response

The article concerns the topic of microRNAs which regulate human and mouse naïve to primed transition of pluripotent stem cells as well as diapause stage of this transition. This paper is a meta-analysis of micro-RNA-seq, RNA-seq and metabolomics datasets form human and mouse pluripotent cells. In my opinion the results presented in this article are not novel, however it is a good database for scientist working with pluripotent stem cells.

As I understand, the meta-analysis does not need additional results to confirm the high throughput data. However, if authors decided to add some of their own results in Fig. 1 thus I must suggest to add also other crucial results to confirm the stated data: activation of Hedgehog signaling pathway (maybe on/off mechanism), functional assessment of metabolism (LDH activity, OXPHOS activity, glutamate assay).

Response: Shh  is an important regulator of embryonic development, however  it has been shown to be minimally active in pluripotent stem cells (Wu et al, 2010), we now summarize those data from the literature in fig2A. We have tried using the pGL3-8XGliRE-ffluc reporter construct in naïve and primed hESC, however the results after transient transfection of the reporter were very variable. It could be due to the low level of shh activity in naïve and primed ESC or to the low level of transfection in those lines. A stable ESC line expressing the GLiRE reporter is  necessary to reliably assess the shh activity (or lack of activity) in ESC, however the generation of such line is beyond the scope of this review.

The metabolism of naïve vs primed mouse and human ESC has been studied by our group and others (Zhou et al 2012, Takashima et al,  2014, Sperber et al 2015, Gu et al 2016, Zhang et al 2016, Sun et al 2018, Baha et al 2018, Cornacchia et al 2019, Mathieu and Ruohola-Baker 2017). In particular, Seahorse experiments have shown that even though primed hESC have more mature mitochondria, OXPHOS activity is decreased (Zhou et al 2012, Sperber et al 2015, Takashima et al  2014). In addition, inhibition of lactate dehydrogenase by oxamate resulted in higher extra-acidification rate (ECAR) and lower oxygen consumption rate (OCR) in primed ESC compared to naïve ESC (Zhou et al 2012). It has also been shown that both naïve and primed ESC rely on glucose as a source of energy (Gu et al, 2016). In addition, we have shown that while fatty acid synthesis is higher in primed , only naïve ESC can use fatty acid oxidation, resulting in accumulation of fatty acids in primed mESC and hESC (Sperber et al 2015). Those functional data are now discussed and summarized in Table 2.

In my opinion, the manuscript is clearly written and easy to understand. However I found some mistakes and phrases which need explanation:

Line 20 - Hh-pathway – explain the abbreviation

Line 31 - Start the sentence with a capital letter (please correct throughout the manuscript)

Line 31 (and throughout the manuscript) – Please add the space before the citation and delete after

Line 65 – it should not be written in italics

Fig. 2C and F – The figures are too small

Fig. 2G – Description is for fig. G, however on the figure there is no “G” but “H”

Fig. 4 - Figure is too big comparing to others. The description: Figure 4, should be deleted from above the Figure.

Response: We thank the reviewer for pointing those out. We have now corrected the mistakes and modified the figures.

Methodology section – the more  detailed description of high throughput data should be added. Also the GSE numbers from GEO database of all presented results should be added.

Response: We followed the reviewer’s suggestion and added the accession numbers for the datasets we used in Table 1. Table 1 now provided a comprehensive description of the public dataset used in this study, including data type, description of experimental condition, culture media, and GEO accession number.

The text is well written, presented figures fulfills the raised topic. They are also well described and discussed. The reference are well cited. In my opinion manuscript should be accepted for publication after minor revision.

Reviewer 3 Report

In the manuscript by Wang, et al, the authors used parallel analysis of microRNA, transcriptomics and metabolomic profiles of human and mouse pluripotency states from previously published datasets to characterize potential roles of miRNAs in pluripotency. They identified 2184 microRNA- target gene connections that are consistently differentially expressed between the naïve and primed state.

The authors then focus on the Hedgehog pathway which is known to be minimally important for maintenance of pluripotency. In a previous paper, the authors identified a ciliary G-protein coupled receptor, GPR161, as a regulator of naïve to primed transition in hESCs. Here, they generated a GPR161 KO hESC line and found an upregulation of one of the Hh pathway components, SMO, in primed KO cells by qPCR. Using RNA seq analysis they mention, but do not provide data, suggesting upregulation of the Gli family of transcription factors which are known targets of the Hh pathway.

Additionally, the authors’ comparison of naïve and primed hESCs datasets confirmed that some of the components of polyamine pathway, like ODC1 and spermidine, are upregulated in naïve hESCs as compared to the primed state. ODC1 and spermidine are established regulators of naïve mouse ESC self-renewal and the authors suggest similar roles in naïve human ESCs.

The authors continue to  analyze microRNAs to obtain a list of 115 common microRNAs in mouse and human pluripotency. Interestingly, their analysis resulted in only 6 microRNAs that were consistently changed across three different mouse microRNA studies and no enrichment of upregulated target-genes in any significant GO terms.

The authors finally focused on the diapause state and compared microarray and RNA seq data of diapause and re-activated/post-implantation embryos to identify 379 consistent connections between 38 microRNAs and the target-genes. The upregulated target genes show an enrichment in stress and hormone response, on the other hand, downregulated genes show an enrichment in cellular morphogenesis and development, consistent with the diapause phenotype. However, the RNA seq data used in this analysis is from a paper under review.

In summary, this manuscript has several critical flaws and provides little new insight into miRNAs and pluripotency.

Major Issues-

The manuscript does not adequately address the different types of analyses used in each previously published dataset and how it could affect their results. None of the results mention the expression levels of microRNAs. This is important since microRNAs that are very lowly expressed are not likely to be present at high enough concentration to have a significant role post-transcriptional gene regulation. The manuscript does not include any signficant functional analysis with the GPR161 KO hESC line. For example, is cilia formation affected in KO primed hESCs? Why isn’t the RNA seq data analysis of WT and KO line included in this manuscript. The manuscript focused on the Hedgehog pathway which is known to be minimally important in the regulation of pluripotency states which weakens the overall impact of the paper. The manuscript identified only 6 microRNAs that consistently changed between three different mouse microRNA studies which undermines their approach to use previously published data. The manuscript does not consider that many microRNAs that are differentially expressed between the two states are functionally redundant and can target the same genes (i.e. miR-290/302). Hence, the authors should delineate between microRNA seed sequences. In the list of upregulated microRNAs in diapause state, miR-290 is reported to have no consistent change between naïve and primed pluripotency. This is one example of a family that has overwhelming data suggesting differential expression, thus lending doubt to the quality of data used in the manuscript. The use of data from a paper under review elsewhere presents a barrier to evaluation of the diapause work. The manuscript lacks a significant summary/conclusion.

Minor issues-

The manuscript is sloppy and has many typos and errors. Below are a few examples. Line #’s listed with corrections in parentheses.

29  intermediate, “the” embryonic diapause stage. 33 translational repression or transcript destabilization[1] . “microRNAs” have been shown to be essential 38 genes at the same time, and multiple microRNAs can target the same gene. “microRNAs” have 47 “microRNAs”[21-24] , including miR-302 cluster, 65 mESC[37-39] , and Dgcr8 KO showed that “microRNAs” are essential for the 66 primed mEpiSC[38] . In particular, the miR-302 cluster is expressed at higher “levels” in mEpiSC 69 “microRNAs” in naïve and primed human pluripotent stem cells. 72 cluster[22] , suggesting common “microRNAs are” involved in mouse and 73 transition. In this paper”,” we compared naïve hESC grown 86 “microRNAs” can control Sonic Hedgehog (shh) activity in human pluripotent cells. Indeed, “microRNAs” 96 significantly differentially expressed between human naïve and primed “ESCs” (Table S1). 161 Figure 2. Hh pathway in naïve-to-primed transition. A. “Expression” of shh pathway components in 167 premature STOP codon. E. GPR161 immunofluorescence staining in Elf1 WT “and” Elf1 GPR161 mutant 168 grown in primed conditions (TeSR). F. ”Shh” component SMO is up-regulated in Elf1 GPR161 mutant 170 naïve and primed hESC through “microRNAs” 175 its target genes in fatty acid activation (ACSL1), synthesis (FASN), elongation (ELOVL7) “and” 313 microRNAs with lower expression “in” the diapause state compared to reactivated state. MicroRNAs that 318 Top 10 microRNAs with highest expression “in” the diapause state compared to reactivated state. 347 cells were cultured at 37 “degrees Celsius”, 5% CO2 and 5% O2. 355 AAAACGCGACAGGTGAGAGG) and “sent” for Sanger sequencing.

Author Response

In the manuscript by Wang, et al, the authors used parallel analysis of microRNA, transcriptomics and metabolomic profiles of human and mouse pluripotency states from previously published datasets to characterize potential roles of miRNAs in pluripotency. They identified 2184 microRNA- target gene connections that are consistently differentially expressed between the naïve and primed state.

The authors then focus on the Hedgehog pathway which is known to be minimally important for maintenance of pluripotency. In a previous paper, the authors identified a ciliary G-protein coupled receptor, GPR161, as a regulator of naïve to primed transition in hESCs. Here, they generated a GPR161 KO hESC line and found an upregulation of one of the Hh pathway components, SMO, in primed KO cells by qPCR. Using RNA seq analysis they mention, but do not provide data, suggesting upregulation of the Gli family of transcription factors which are known targets of the Hh pathway.

Additionally, the authors’ comparison of naïve and primed hESCs datasets confirmed that some of the components of polyamine pathway, like ODC1 and spermidine, are upregulated in naïve hESCs as compared to the primed state. ODC1 and spermidine are established regulators of naïve mouse ESC self-renewal and the authors suggest similar roles in naïve human ESCs.

The authors continue to  analyze microRNAs to obtain a list of 115 common microRNAs in mouse and human pluripotency. Interestingly, their analysis resulted in only 6 microRNAs that were consistently changed across three different mouse microRNA studies and no enrichment of upregulated target-genes in any significant GO terms.

The authors finally focused on the diapause state and compared microarray and RNA seq data of diapause and re-activated/post-implantation embryos to identify 379 consistent connections between 38 microRNAs and the target-genes. The upregulated target genes show an enrichment in stress and hormone response, on the other hand, downregulated genes show an enrichment in cellular morphogenesis and development, consistent with the diapause phenotype. However, the RNA seq data used in this analysis is from a paper under review.

Response: The paper from which we obtained the diapause RNA-seq data is now accepted for publication at Developmental Cell, and the source dataset will be made publicly available upon publication. The detail analysis and visualization of the diapause RNA-seq dataset is available in the link below:

https://homes.cs.washington.edu/~ruzzo/papers/Margaretha/test.html

In summary, this manuscript has several critical flaws and provides little new insight into miRNAs and pluripotency.

Major Issues-

The manuscript does not adequately address the different types of analyses used in each previously published dataset and how it could affect their results. None of the results mention the expression levels of microRNAs. This is important since microRNAs that are very lowly expressed are not likely to be present at high enough concentration to have a significant role post-transcriptional gene regulation.

Response: We agree with the reviewer about the importance of absolute levels of microRNA expression and now included in supplemental tables, in addition to the fold changes presented in existing tables. Table S1 included absolute expression levels (normalized read counts) of 357 microRNAs differentially expressed between human naïve and primed states. Table S5 included normalized read counts of 115 microRNAs that are changed in both human and mouse naïve to primed transition.

The manuscript does not include any significant functional analysis with the GPR161 KO hESC line. For example, is cilia formation affected in KO primed hESCs? Why isn’t the RNA seq data analysis of WT and KO line included in this manuscript. The manuscript focused on the Hedgehog pathway which is known to be minimally important in the regulation of pluripotency states which weakens the overall impact of the paper.

Response: We have now performed more analysis of the GPR161 mutant hESC line. We found that cilia formation is not affected in the mutant cells (Fig2F). We added a panel (Fig2A) summarizing the literature on shh activity during embryonic development. Even though shh is minimally active in ESC, the mechanisms repressing this pathway is not well understood. We propose that cilia formation, GPR161 and microRNA could play a role in this repression. In particular, we propose that shh pathway could be regulated differently in naïve and primed hESC. We agree with the reviewer that more functional data is required to validate this hypothesis, however in this manuscript our goal is to bring this hypothesis to the field.

The manuscript identified only 6 microRNAs that consistently changed between three different mouse microRNA studies which undermines their approach to use previously published data. The manuscript does not consider that many microRNAs that are differentially expressed between the two states are functionally redundant and can target the same genes (i.e. miR-290/302). Hence, the authors should delineate between microRNA seed sequences. In the list of upregulated microRNAs in diapause state, miR-290 is reported to have no consistent change between naïve and primed pluripotency. This is one example of a family that has overwhelming data suggesting differential expression, thus lending doubt to the quality of data used in the manuscript.

Response: We found that many microRNAs are not found as consistently changing because of the -3p/-5p distinction at the end of microRNA IDs. For example, when previous studies established the role of miR-290 in  pluripotency, they did not distinguish miR-290-3p vs. miR-290-5p (Gu et al, Cell Research 26:350–366 (2016)). We combined the -3p/-5p forms, re-analyzed the three mouse microRNA-seq datasets and found that the number of consistently changing microRNAs (>1.5 fold change in two out of three studies) increased from 131 in the original analysis to 221 in the new analysis. More importantly, miR-290 is now classified as consistently changing. The number of microRNAs that significantly changed in all three datasets increased from 6 to 8

The use of data from a paper under review elsewhere presents a barrier to evaluation of the diapause work.

Response: The paper from which we obtained the diapause RNA-seq data is now accepted for publication at Developmental Cell, and the source dataset will be made publicly available upon publication. The detail analysis and visualization of the diapause RNA-seq dataset is available in the link below:

https://homes.cs.washington.edu/~ruzzo/papers/Margaretha/test.html

The manuscript lacks a significant summary/conclusion.

We have now modified the conclusion section (#4) based on reviewer’s suggestion.

Minor issues-

The manuscript is sloppy and has many typos and errors. Below are a few examples. Line #’s listed with corrections in parentheses.

29  intermediate, “the” embryonic diapause stage. 33 translational repression or transcript destabilization[1] . “microRNAs” have been shown to be essential 38 genes at the same time, and multiple microRNAs can target the same gene. “microRNAs” have 47 “microRNAs”[21-24] , including miR-302 cluster, 65 mESC[37-39] , and Dgcr8 KO showed that “microRNAs” are essential for the 66 primed mEpiSC[38] . In particular, the miR-302 cluster is expressed at higher “levels” in mEpiSC 69 “microRNAs” in naïve and primed human pluripotent stem cells. 72 cluster[22] , suggesting common “microRNAs are” involved in mouse and 73 transition. In this paper”,” we compared naïve hESC grown 86 “microRNAs” can control Sonic Hedgehog (shh) activity in human pluripotent cells. Indeed, “microRNAs” 96 significantly differentially expressed between human naïve and primed “ESCs” (Table S1). 161 Figure 2. Hh pathway in naïve-to-primed transition. A. “Expression” of shh pathway components in 167 premature STOP codon. E. GPR161 immunofluorescence staining in Elf1 WT “and” Elf1 GPR161 mutant 168 grown in primed conditions (TeSR). F. ”Shh” component SMO is up-regulated in Elf1 GPR161 mutant 170 naïve and primed hESC through “microRNAs” 175 its target genes in fatty acid activation (ACSL1), synthesis (FASN), elongation (ELOVL7) “and” 313 microRNAs with lower expression “in” the diapause state compared to reactivated state. MicroRNAs that 318 Top 10 microRNAs with highest expression “in” the diapause state compared to reactivated state. 347 cells were cultured at 37 “degrees Celsius”, 5% CO2 and 5% O2. 355 AAAACGCGACAGGTGAGAGG) and “sent” for Sanger sequencing.

Response: We thank the reviewer for pointing out those mistakes. We have now corrected these typos and errors.

Round 2

Reviewer 3 Report

In the revised manuscript, Wang et al, address concerns regarding inadequate functional analysis of the GPR161 KO line by demonstrating no loss of cilia formation in the primed state. Importantly, the authors fail to provide evidence of misregulation of PKA (they now mention it should) in the GPR161 KO line. Besides upregulation of Smo, does this cell line have any phenotype? Some sort of quantification of the 3D re-constructed view of Arl13/ZO-1 staining would strengthen the point of no defect in cilia formation. Justification for focusing on the hedgehog pathway is still relatively thin as it is not relevant in the cell states being studied. The graphical representation of their hypothesized model in fig2I is not viewable. Why not differentiate into a cell type that utilizes Hh signaling and show a defect? Alternatively, the authors could add more description/hypothesis of how Hh suppression might be regulated in naïve vs primed and why this is relevant as part of the discussion. Finally, the authors add data to address absolute expression levels and re-analyze the data to be more consistent with known changes in miRNA expression.

Minor point:

Line-209 should read miR-17~92.

With minor updates the manuscript is acceptable for publication.

Author Response

In the revised manuscript, Wang et al, address concerns regarding inadequate functional analysis of the GPR161 KO line by demonstrating no loss of cilia formation in the primed state. Importantly, the authors fail to provide evidence of misregulation of PKA (they now mention it should) in the GPR161 KO line. Besides upregulation of Smo, does this cell line have any phenotype? Some sort of quantification of the 3D re-constructed view of Arl13/ZO-1 staining would strengthen the point of no defect in cilia formation. Justification for focusing on the hedgehog pathway is still relatively thin as it is not relevant in the cell states being studied. The graphical representation of their hypothesized model in fig2I is not viewable. Why not differentiate into a cell type that utilizes Hh signaling and show a defect? Alternatively, the authors could add more description/hypothesis of how Hh suppression might be regulated in naïve vs primed and why this is relevant as part of the discussion. Finally, the authors add data to address absolute expression levels and re-analyze the data to be more consistent with known changes in miRNA expression.

Response: We thank the reviewer for these comments. We agree with the reviewer that the exciting GPR161 KO studies warrant further experiments and analysis.  Our preliminary analysis actually shows three times more cilia in GPR161 KO than in control. We are very excited this potential new role of GPR161 for determining the number of cilia in iPSC.  However, many more follow up experiments are required to understand the mechanism. The experiments are underway and they will be published in a follow up manuscript. We thank the reviewer for pointing out the interest in this topic. These preliminary data further culminate the importance for Hh pathway repression in stem cell stages.  Further studies will reveal the mechanism for iPSC to accomplish this repressive action.

As suggested by the reviewer, we have now added more description/hypothesis of how we propose Hh suppression is established in naïve and primed hESC, to maintain their pluripotent stage. This can be found in the section #4.Conclusions.

Minor point:

Line-209 should read miR-17~92.

Response: We have now made this correction.

With minor updates the manuscript is acceptable for publication.

Response: Thank you.